# Detection of *Mannheimia haemolytica*-Specific IgG, IgM and IgA in Sera and Their Relationship to Respiratory Disease in Cattle

**DOI:** 10.3390/ani13091531

**Published:** 2023-05-03

**Authors:** Korakrit Poonsuk, Carita Kordik, Matthew Hille, Ting-Yu Cheng, William B. Crosby, Amelia R. Woolums, Michael L. Clawson, Carol Chitko-McKown, Bruce Brodersen, John Dustin Loy

**Affiliations:** 1Nebraska Veterinary Diagnostic Center, School of Veterinary Medicine and Biomedical Sciences, University of Nebraska–Lincoln, Lincoln, NE 68503, USA; ckordik2@unl.edu (C.K.); mhille@unl.edu (M.H.); bbrodersen1@unl.edu (B.B.); jdloy@unl.edu (J.D.L.); 2Department of Veterinary Preventive Medicine, College of Veterinary Medicine, The Ohio State University, Columbus, OH 43210, USA; cheng.1784@osu.edu; 3Department of Pathobiology and Population Medicine, Mississippi State University, Mississippi State, MS 39762, USA; wbc95@msstate.edu (W.B.C.); amelia.woolums@msstate.edu (A.R.W.); 4United States Department of Agriculture (USDA), Agricultural Research Service (ARS), United States Meat Animal Research Center, Clay Center, NE 68933, USA; mike.clawson@usda.gov (M.L.C.); carol.chitkomckown@usda.gov (C.C.-M.)

**Keywords:** *Mannheimia haemolytica*, bovine respiratory disease, antibody, ELISA, IgG, IgM, IgA

## Abstract

**Simple Summary:**

*Mannheimia haemolytica* is a key contributor to bovine respiratory disease (BRD) in cattle. It is commonly found in calves and young cattle with enzootic pneumonia. Traditional visual assessments for BRD can be unreliable, leading to improper treatment and disease spread. The indirect ELISAs described in this study detect specific *M. haemolytica*-IgG, IgM, and IgA, and have demonstrated high diagnostic sensitivity and specificity. These tests can be used for disease surveillance and prevention in feedlot cattle.

**Abstract:**

*Mannheimia haemolytica* is one of the major causes of bovine respiratory disease in cattle. The organism is the primary bacterium isolated from calves and young cattle affected with enzootic pneumonia. Novel indirect ELISAs were developed and evaluated to enable quantification of antibody responses to whole cell antigens using *M. haemolytica* A1 strain P1148. In this study, the ELISAs were initially developed using sera from both *M. haemolytica*-culture-free and clinically infected cattle, then the final prototypes were tested in the validation phase using a larger set of known-status *M. haemolytica* sera (*n* = 145) collected from feedlot cattle. The test showed good inter-assay and intra-assay repeatability. Diagnostic sensitivity and specificity were estimated at 91% and 87% for IgG at a cutoff of S/P ≥ 0.8. IgM diagnostic sensitivity and specificity were 91% and 81% at a cutoff of sample to positive (S/P) ratio ≥ 0.8. IgA diagnostic sensitivity was 89% whereas specificity was 78% at a cutoff of S/P ≥ 0.2. ELISA results of all isotypes were related to the diagnosis of respiratory disease and isolation of *M. haemolytica* (*p*-value < 0.05). These data suggest that *M. haemolytica* ELISAs can be adapted to the detection and quantification of antibody in serum specimens and support the use of these tests for the disease surveillance and disease prevention research in feedlot cattle.

## 1. Introduction

Bovine respiratory disease (BRD), can be caused by multiple viral and bacterial agents together with stressors and environmental factors. BRD is one of the most important diseases of cattle, responsible for up to 70–80% morbidity and 40–50% of mortality in feedlot cattle [1,2,3]. Economic losses to the North American feedlot industry due to BRD have been estimated to be as high as USD one billion annually [4,5,6]. Among these, *Mannheimia haemolytica* has been the predominant pathogenic agent, detected in up to 91% of clinically affected cattle [3,7,8,9]. *M. haemolytica* is a Gram-negative, non-motile, non-spore-forming, facultative anaerobic coccobacilli [10]. The organism is a significant component of enzootic pneumonia in neonatal calves and frequently isolated from feedlot cattle suffering from BRD [11,12,13]. Although the organism naturally exists as a commensal of the upper respiratory tract and nasopharynx of healthy cattle [14,15] the organism is considered an opportunistic and economically important pathogen to the global cattle industry, especially in the North America.

BRD caused by *M. haemolytica* is considered a management disease, resulting from an incompatibility between the biology of calves, the pathogen, and management systems that may cause stress [16]. Vaccines and antibiotics have been extensively used as part of the practices to control and reduce losses due to the disease, alongside with improved management strategies [7]. However, despite control and treatment efforts, the key for successful management of BRD in feedlot cattle relies on the ability to identify cattle at risk for contracting the disease, combined with rapid detection and identification of the causative agents to guide efficacious treatment methods. Distinguishing among etiologic agents of BRD based upon clinical signs is generally unsuccessful [17]. Isolation of causative agents in post-mortem tissue samples is commonly performed; however, the utility of post-mortem confirmation is limited in practice [18]. Ante-mortem diagnostic results early in the disease process promotes the timely treatment and management modification to control the disease transmission and reduce the losses in affected animals [19,20]. Serological testing for *M. haemolytica* would allow for the detection of antibodies that can be used to confirm prior exposure and levels of humoral immune responses to the bacteria. Overall, serological testing for *M. haemolytica* can provide valuable information for animal health management and disease control. There are several reasons why serological testing for *M. haemolytica* is could be useful, e.g., (1) Diagnosis: Serological testing can be used to diagnose an active or past infection with *M. haemolytica*. It can also be used to confirm a clinical diagnosis of pneumonia in affected cattle and identification of higher risk animals; (2) Epidemiology: Serological testing can be used to determine the prevalence of *M. haemolytica* infection in a herd or flock. This information can be used to understand the spread of the bacteria and the risk factors associated with infection; (3) Vaccination: Serological testing can be used to evaluate the responses to a vaccine or used in vaccination program evaluation; (4) Surveillance: Serological testing can be used in disease surveillance programs to monitor the emergence of *M. haemolytica* and to detect changes in the prevalence of the bacteria indirectly through immune responses [21,22,23]. 

Immunoglobulin isotypes are different forms of the immunoglobulin (Ig) protein that have distinct structural and functional characteristics [24]. The main immunoglobulin isotypes that contribute to the immune response against infectious diseases are IgM, IgG, and IgA. In general, IgM is the first immunoglobulin to be produced during an acute infection of *M. haemolytica* [25]. Its pentameric structure allows for a high binding capacity and a high avidity for antigens. It is also effective at activating the complement system, which leads to the lysis of bacteria and the recruitment of immune cells to the site of infection. IgM is the primary antibody that is responsible for the opsonization of bacteria, which enhances their clearance by phagocytes [26,27]. IgG is produced later in the immune response, after IgM [25]. It is a monomeric immunoglobulin, and the most abundant immunoglobulin in the blood. As it has the longest half-life of all the isotypes, IgG is important for providing long-term immunity against pathogens, as it can persist in the body for months or even years after the initial infection. It is also effective at neutralizing pathogens and activating the complement system [28]. IgG is concentrated in colostrum providing protection to the neonate, which is important in preventing early infections in calves [29]. IgA is the second most abundant immunoglobulin in the body and is found in high concentrations in mucosal surfaces such as the gut, respiratory tract, and genitourinary tract, and prevents pathogens from establishing an infection at these sites [30]. That is, IgA is likely an important immunoglobulin in the immune response against *M. haemolytica* infections, particularly along the mucosal surfaces of the respiratory tract [25]. IgA can neutralize the bacteria, activate the complement system, and bind to antigens in a non-inflammatory manner, which helps to prevent excessive inflammation and tissue damage [31]. 

The antibody responses to *M. haemolytica* remain an area of ongoing research. Studies have shown that both IgG and IgA isotypes are produced in response to *M. haemolytica* infection, with IgG being the dominant isotype [25,32,33]. However, the exact role of these antibodies in protecting against BRD is not fully understood. Some studies have suggested that IgG antibodies may play a role in neutralizing toxins produced by *M. haemolytica* and in opsonizing the bacteria, which can enhance phagocytosis by white blood cells [34,35]. However, other studies have found that the level of IgG antibodies does not always correlate with protection against BRD, suggesting that other factors may also be involved [36,37]. IgA antibodies have been found in respiratory secretions of infected animals, suggesting that they may play a role in protecting the respiratory tract from *M. haemolytica* colonization [25,38]. However, the specific mechanisms by which IgA may exert this protection have not been studied. Overall, more research is needed to fully understand the role of antibodies in protecting against *M. haemolytica* infection and the development of BRD. The use of different vaccination protocols, genetics of the host, and environmental factors may influence the antibody response. Isotype-specific serological tests can be used to detect the presence and levels of specific immunoglobulin isotypes, such as IgM, IgG, and IgA, in response to a *M. haemolytica* infection. 

The present study aimed to develop three serodiagnostic assays for *M. haemolytica*-specific IgG, IgM, and IgA detection. Indirect ELISAs (enzyme-linked immunosorbent assays) were developed and evaluated to enable quantification of antibody responses to whole cell antigen using *M. haemolytica* A1 strain P1148 (ATCC 14003). Previous work has utilized a whole cell ELISA to study antibody responses to *M. haemolytica*, however, our goal was to enhance this method using advanced blocking approaches developed for a similar bovine opportunistic pathogen, *Moraxella bovis* [39], and also to examine specific responses of different Ig types. We evaluated relationships between the serum antibody levels and *M. haemolytica* susceptibility in feedlot cattle using these tests. Importantly, these serotype-specific *M. haemolytica* ELISAs can be used in different applications, i.e., (1) Assessing the stage of the infection: IgM is typically the first immunoglobulin to be produced during an acute infection, whereas IgG is produced later in the immune response. By measuring the levels of IgM and IgG, a serological test may provide information about the stage of the infection, whether it is acute or chronic; (2) Measuring the immune response: IgM, IgG, and IgA have different functions and roles in the immune response against *M. haemolytica*. By measuring the levels of these isotypes, a serological test may provide information about the strength and effectiveness of the immune response; (3) Monitoring the effectiveness of treatment: Serological tests can be used to monitor the effectiveness of treatment for *M. haemolytica* infections. By measuring changes in the levels of specific immunoglobulin isotypes over time, a serological test can provide information about whether the treatment is working and if the infection is being cleared; (4) Identifying potential risk factors: Serological tests can be used to identify potential risk factors for *M. haemolytica* infections [25,32,33]. By measuring the levels of specific immunoglobulin isotypes in different populations or groups, a serological test can provide information about which individuals or groups are at higher risk for infection; (5) Understanding the infection: Serological tests can be used to study the role of specific immunoglobulin isotypes in the immune response to *M. haemolytica*, which can help in developing new treatments and vaccines [22,40,41]. In summary, isotype-specific serological tests for *M. haemolytica* are potentially important for assessing the stage of the infection, measuring the immune response, monitoring the effectiveness of treatment, identifying potential risk factors, and better understanding the disease ecology.

## 2. Materials and Methods

### 2.1. Study Design

The objective of this study was to develop IgG, IgM, and IgA isotype-specific ELISAs for the detection of *M. haemolytica*-specific antibodies in cattle sera using *M. haemolytica* A1 strain P1148 as an antigen. The study was divided into two parts: test development and test validation. In the test development phase, different prototypes of the ELISAs were developed using a set of serum or plasma specimens obtained from both *M. haemolytica*-culture-free and *M. haemolytica* clinically infected animals. The conditions for the prototypes varied in terms of reagents, incubation temperature, and incubation time at each step. The IgG, IgM, and IgA isotype-specific *M. haemolytica* ELISA were developed using a carefully selected set of serum specimens obtained from both animals free of disease attributable to *M. haemolytica* and *M. haemolytica* clinically infected animals. The *M. haemolytica* antibody-negative serum samples were collected from 46 calves that participated in an infectious bovine keratoconjunctivitis vaccine trial, while the clinically infected, *M. haemolytica* antibody-positive plasma samples were collected from 5 cattle that were diagnosed with BRD and had *M. haemolytica* isolated from lung specimens. 

In the test validation phase, the final prototypes were tested with a larger set of known-status *M. haemolytica* sera obtained from a larger field trial. Serum samples were collected from stocker cattle at the day of arrival (study day 0) and the last day of the experiment (study day 20 or 21). The serum samples from cattle clinically normal and negative to *M. haemolytica* culture at study day 0 were used as antibody negative specimens. All serum samples collected from cattle which were culture-positive for *M. haemolytica* at least 14 days after the culture-positive sampling date were used as true antibody positive specimens for test validation. Repeatability of the test was then determined using known and unknown *M. haemolytica* antibody status sera.

### 2.2. M. haemolytica A1 Strain P1148 Culture

To prepare the *M. haemolytica* culture, a frozen stock culture of *M. haemolytica* (ATCC #14003) was thawed and streaked onto Tryptic Soy Agar (TSA) supplemented with 5% sheep blood (Remel, Lenexa, KS). The plates were then incubated in an atmosphere of 5% CO_2_ at 37 °C for 24 h. A single colony was selected and transferred to a 50 mL conical tube containing 5 mL of Brain Heart Infusion broth (BHI) (Becton, Dickinson and Company, Sparks, MD, USA) at room temperature. The culture was incubated in a shaking incubator at 37 °C and at a shaking frequency of 200 rpm for 4–6 h. Following the incubation, 1 mL of the culture was transferred to 1 L of BHI broth in a 2 L Erlenmeyer flask and incubated overnight at 37 °C and at a shaking frequency of 90 rpm. The culture was then aliquoted into 50 mL sterile plastic conical tubes, washed three times by centrifugation at 6000× *g* for 15 min, and resuspended in Phosphate Buffered Saline (PBS) (Fisher Scientific, Waltham, MA, USA). After the third wash, the bacterial pellets were suspended in 3 mL of PBS. Ten-fold dilutions were prepared and plated onto TSA agar plates supplemented with 5% sheep blood. The plates were incubated overnight at 37 °C. The number of colony forming units (CFU) in the stock culture were determined by counting the number of colonies formed on the plates. The solutions were diluted according to the determined CFU with 0.4% PBS formalin to reach a final concentration of 10^8^ CFU/100 µL and stored at −80 °C. 

### 2.3. Development of M. haemolytica IgG, IgM, and IgA Isotype-Specific Indirect ELISAs

The IgG, IgM, and IgA isotype-specific *M. haemolytica* ELISAs were developed using a set of serum specimens obtained from both *M. haemolytica*-culture-free and *M. haemolytica* clinically infected animals. The *M. haemolytica* antibody-negative serum samples were collected from 46 calves that participated in an infectious bovine keratoconjunctivitis vaccine trial [39]. The clinically infected, *M. haemolytica* antibody-positive plasma samples were collected from 5 cattle that were submitted to the Nebraska Veterinary Diagnostic Center (NVDC) as part of another animal experiment conducted under Kansas State University (IACUC# 3338). These cattle were diagnosed with BRD and had genotype 2 *M. haemolytica* isolated from their lung specimens. 

The ELISAs were optimized using different plates and reagents under various conditions in order to differentiate between negative and positive signals. For the final prototypes, the *M. haemolytica* A1 strain P1148 (10^8^ CFU/100 µL) was diluted 1:50 in PBS with 0.05% sodium azide solution. One hundred microliters of the coating solution was added to 96-well polystyrene microplates (Thermo Fisher Scientific™, Waltham, MA, USA), sealed, and incubated at 4 °C for 18 h. After the incubation, the plate was washed 3 times with deionized water. The coated plate was then blocked using ChonBlock blocking/sample dilution ELISA buffer (Chondrex Inc, Woodinville, WA, USA) according to the manufacturer’s recommendations. After the blocking step, the plate was washed with TRIS-buffered saline with 2% Tween 20 (TBS-Tween20). The plate was dried and kept in resealable foil pouch with desiccants at 4 °C until used.

The testing procedures are different based on immunoglobulin isotype. In brief, serum samples were diluted 1:100 with ChonBlock blocking/sample dilution ELISA buffer for IgG and IgM and diluted 1:50 for IgA isotype-specific *M. haemolytica* ELISAs in 96-well polystyrene pre-dilution microplate then transferred to the ELISA plate. The plate was sealed and incubated at 37 °C for one hour. After incubation, the plate was washed with TBS-Tween20. Isotype-specific secondary antibody conjugated to horseradish peroxidase (Bethyl Laboratories, Inc.) was added to the plate and then incubated at room temperature (IgG and IgM) or 37 °C (IgA) for one hour. Reactions were developed using TMB (3,3′,5,5′-Tetramethylbenzidine) substrate (Alpha Diagnostic Intl Inc., San Antonio, TX, USA) for 10 min, stopped using a stop solution, and read at 450 nm using a microplate spectrophotometer. The total incubation time for the assay, including all steps, is approximately 2 h and 10 min.

The results were represented as sample-to-positive (S/P ratio calculated using the formulation below:(1)$$S/P\;ratio=\frac{\left(sample\;OD-blank\;well\;control\;mean\;OD\right)}{\left(positive\;control\;mean\;OD-blank\;well\;control\;mean\;OD\right)}$$

### 2.4. Study Population and Sampling for the Test Validation

Specimens used for the *M. haemolytica* IgG, IgM, and IgA isotype-specific test validation were obtained from an experiment determining the effects of tulathromycin metaphylaxis on BRD incidence in stocker cattle. This study was reviewed and approved by the Mississippi State University Institutional Animal Care and Use Committee under Protocols IACUC-18-529 and IACUC-21-558. In brief, the study was conducted in 4 trials with a total of approximately 80 animals per trial. The trials started in late October (Fall 2019, 2020, and 2021) and in mid-March (Spring 2021). Cattle were obtained from regional auction markets near Starkville, MS and arrived at the study site within 6 h of transport. On arrival, animals were weighed, had their rectal temperature recorded, and underwent day 0 sampling. They were tagged with individual IDs and color-coded tags according to their groups. The animals were randomly assigned to either receive tulathromycin metaphylaxis (META) or not (NO META) and were separated into two 40-acre pastures with no fence-line contact. The META group received tulathromycin on day 0. Cattle in both groups were monitored for BRD daily by trained pen riders on horseback for 20–21 days. Animals received respiratory and clostridial vaccines and were dewormed with fenbendazole and doramectin according to label directions. Ear notch samples were collected for bovine viral diarrhea virus testing and animals found to be persistently infected were removed from the study and euthanized.

Blood samples were collected via jugular venipuncture on days 0, 7, 14, and 20 or 21 and serum was harvested and stored at −80 °C until further use. The animals were monitored daily for clinical BRD using criteria described elsewhere [42], i.e., animals with BRD score 1 or 2 with rectal temperature ≥ 104 °F or a BRD score ≥ 3 regardless of rectal temperature and no signs of other diseases were classified as BRD-affected cases. In the event of respiratory disease, the cattle were treated with a regimen of ceftiofur, florfenicol, or oxytetracycline. Nasopharyngeal swabs were collected from affected cattle, and the causative agents, including *M. haemolytica*, were determined using methods described elsewhere [42]. Serum specimens collected during the study were classified into *M. haemolytica* positive or negative status according to parameters outlined below.

### 2.5. Test Validation

The performance of the IgG, IgM, and IgA isotype-specific *M. haemolytica* ELISAs was validated using known-status serum specimens obtained from the aforementioned animal experiment. The serum samples used for validation were obtained from cattle that were either culture-negative or culture-positive for *M. haemolytica*. That is, for the validation of the IgG ELISA, serum samples collected on study day 0 from the cattle that were negative for *M. haemolytica* culture (*n* = 100) were used as antibody negative specimens. All serum samples collected from cattle that were culture-positive for *M. haemolytica* at least 14 days after the culture-positive sampling date (*n* = 45) were used as antibody positive specimens. The IgM and IgA ELISAs were validated using the same set of samples thereafter.

In order to ensure the accuracy of the assays, antibody negative and positive samples from the test development step were included in duplicate in each assay plate as negative and positive controls. The inclusion of negative and positive controls in the assay plates helped to confirm the specificity and sensitivity of the assays.

### 2.6. Test Repeatability Determination

Inter-assay repeatability was determined for each developed ELISA by testing a set of samples with known and unknown *M. haemolytica* antibody levels. A total of 30 samples including 5 with high antibody levels, 5 with low antibody levels, and 20 unknown samples were randomly selected from the test development trial and repeatedly tested for 4 rounds to estimate the test repeatability allowing a 10% half-width of the 95% CI of the within-subject standard deviation (Sw) [43]. To avoid possible bias, the samples were aliquoted into 4 separate sets with randomly assigned sample ID. Four technicians at the NVDC were assigned to test one sample set at a time, at least 20 h apart, following the same procedure without knowing the status of each sample.

Intra-assay repeatability was determined by testing a set of samples repeatedly within the same plate. Due to space limitation, a total of 8 samples with different *M. haemolytica*-specific antibody levels were tested in 12 replicates at the same time in 1 plate per assay to provide an estimate with a 20.8% half-width of the 95% CI of Sw. Similar to the inter-assay repeatability, samples were randomly selected from the test development trial, assigned a random sample ID, and tested blindly in random locations within the same plate. Positive and negative controls were included in duplicate in each assay to confirm the validity of the test.

### 2.7. Statistical Analyses

In order to determine the repeatability of the *M. haemolytica* IgG, IgM, and IgA isotype-specific ELISAs, both inter-assay and intra-assay repeatability were assessed by calculating intra-class correlation coefficient (ICC) using the R package (https://cran.r-project.org/web/packages/irr/index.html, accessed on 17 February 2023). Previously established ICC classification guidelines were used to interpret the repeatability: (1) <50% (poor); (2) 50–75% (moderate); (3) 75–90% (good); (4) >90% (excellent) [44].

ROC curve analyses were completed to evaluate the performance of the *M. haemolytica* IgG, IgM, and IgA ELISAs using the R software package *pROC* (https://cran.r-project.org/web/packages/pROC/index.html, accessed on 17 February 2023) [45]. A non-parametric DeLong method was used to estimate the 95% CIs for the area under the curve (AUC) and compare AUCs among isotypes [46]. Diagnostic sensitivity and specificity were derived from the ROC analyses for specific assay cutoffs. To estimate the diagnostic sensitivity and specificity confidence intervals, a nonparametric stratified bootstrapping method with 10,000 iterations was used [45,47]. For each iteration i, a sample (Xi) was created by randomly assigning the S/P ratio in positive and negative samples to one of two strata, each with the same size as the number of “true positive” and “true negative” samples. The diagnostic sensitivity and specificity for specific cutoffs were then calculated for Xi. The confidence interval lower and upper bounds were computed as the 5th and 95th percentiles of the sensitivities or specificities derived from 10,000 iterations [47]. The optimal cutoffs were determined using Youden’s J index [48] based on the balance between sensitivity and specificity. 

The relationship between the isolation of *M. haemolytica* and the detection of IgG, IgM, and IgA against *M. haemolytica* was investigated using three logistical regression models, where each immunoglobulin isotype was separately modeled. The isolation of *M. haemolytica* at the arrival of animals was included in the models as dichotomous outcomes while the S/P ratio of IgG, IgM, and IgA determined by the *M. haemolytica* ELISAs three weeks after the arrival were included as continuous fixed effects. The effect of each isotype was reported as an odds ratio with a profiled likelihood 95% CI, indicating the fold change in odds of *M. haemolytica* isolation at arrival. Significance was declared at *p*-value < 0.05 using the likelihood ratio test, and a trend at 0.05 ≤ *p*-value < 0.10.

## 3. Results

### 3.1. M. haemolytica IgG, IgM, and IgA ELISAs

The *M. haemolytica* IgG, IgM, and IgA isotype-specific ELISAs developed in this study were successful in detecting antibodies against *M. haemolytica*. IgG, IgM, and IgA S/P results are available in Appendix A. Optimized using a selected set of serum specimens from both *M. haemolytica*-culture-free and clinically infected animals, the final prototypes were able to differentiate between negative and positive signals of IgG, IgM, and IgA antibodies specific to the bacteria (Figure 1). The reliability and consistency of the ELISAs in measuring *M. haemolytica*-specific antibody levels were determined by assessing both inter-assay and intra-assay repeatability. Based on the criteria described elsewhere [44], the tests showed good inter-assay repeatability (ICC 87.2–98.6%) and excellent intra-assay repeatability (ICC 99.6–99.9%) (Table 1).

### 3.2. Determination of Specificity and Sensitivity

Similar AUCs were identified for the *M. haemolytica* IgG, IgM, and IgA ELISAs, respectively (DeLong method, *p*-value < 0.001; Figure 2). Three assessed ELISAs were able to correctly differentiate positive and negative samples (AUC > 90%), with true *M. haemolytica* infectious status determined by *M. haemolytica* isolation using samples collected at least 14 days after cultured positive. As determined by Youden’s J index, the optimal cutoff for IgG, IgM, and IgA ELISAs was 0.50. Diagnostic sensitivity and specificity were estimated at 91% and 87% for IgG at a cutoff of S/P ≥ 0.8. IgM diagnostic sensitivity and specificity were 91% and 81% at a cutoff of S/P ≥ 0.8. IgA diagnostic sensitivity was 89% whereas specificity was 78% at a cutoff of S/P ≥ 0.2. Estimated diagnostic sensitivities and specificities by certain cutoffs of each assay are presented in Table 2.

For the logistic regression modeling, a significant relationship was identified between *M. haemolytica* isolation at the arrival of animals and IgG S/P at 3 weeks after the arrival (odds ratio [OR] = 1.58, *p*-value < 0.05). That is, for an average animal, the odds of being isolated with *M. haemolytica* at arrival is estimated to increase by 1.58-fold with each unit increase of IgG S/P 3 weeks after arrival. This positive relationship demonstrated the detection of *M. haemolytica* IgG using our proposed ELISA. On the contrary, a non-significant relationship was found for IgM and IgA ELISAs.

## 4. Discussion

*M. haemolytica* is considered one of the most important pathogens associated with BRD and it is responsible for significant economic losses in the cattle industry worldwide [49,50]. The control of *M. haemolytica* disease is typically based on a combination of management strategies, such as vaccination, antimicrobial treatment, and biosecurity measures [41]. Vaccination is often used to control *M. haemolytica* [51]. However, the effectiveness of vaccination can be influenced by several factors, such as the strain of the pathogen, the timing of vaccination, and the management practices of the herd [16,52]. The ability to accurately assess antibodies specific to *M. haemolytica* using serological tests, such as the ELISAs described here, could help refine the use of vaccines to improve outcomes in vaccinated cattle.

The diagnosis of *M. haemolytica* infection is typically based on a combination of clinical signs and bacteriological culture or PCR to identify the organism. The accurate diagnosis of *M. haemolytica* infection that incorporates a serological component could be important for several reasons: (1) Serological assays can be used to identify the cause that allows for targeted treatment and management strategies; (2) IgG and IgM isotype-specific serological assays could be used for monitoring disease trends within the affected herd, and to track changes in the prevalence and incidence of the disease over time. This could help to monitor the effectiveness of control measures, such as vaccination programs. A combination of different immunoglobulin isotype-specific ELISAs can be used to identify cattle that may be asymptomatic or in the early stages of respiratory infection and thus facilitate proper treatment to mitigate disease spread; (3) Serological testing could improve surveillance and epidemiological studies, to help to understand the transmission dynamics of the disease and also to identify risk factors [22,41]. 

Decades ago, several serological assays were developed for *M. haemolytica* serotype-specific antibody detection including tube agglutination [53,54], double gel diffusion and countercurrent immunoelectrophoresis [55], rapid plate agglutination or slide agglutination [56,57,58,59], indirect hemagglutination [60,61], coagulation test [62], microscopic agglutination test (MAT), and beads-based immunoprecipitation [38]. However, a major problem with traditional serotyping is the presence of untypable isolates [63]. Some of the problems of untypable isolates in the past, before species of *Mannheimia* were classified, were probably also related to diagnostic difficulties which were largely resolved by the classification of *Mannheimia* and its species including *M. haemolytica*, which increased the ability of identification [64]. This assay utilized a serotype A1 *M. haemolytica* strain P1148 as the antigen source. Serotype A1 strain P1148 used in this study is genotype 2, which has been shown to be more frequently associated with disease, and therefore likely to have enhanced utility for an immunoassay [65,66,67]. On the other hand, non-serotype-specific ELISAs have been developed for use in epidemiology research and routine diagnostics [25,38,68,69,70,71,72]. However, no *M. haemolytica* ELISA is currently available or offered as a test in veterinary diagnostic laboratories in the North America. Although our study showed that the ELISAs based on the *M. haemolytica* serotype A1 strain P1148 provided good diagnostic sensitivity and diagnostic specificity, it would be valuable to explore the potential of utilizing additional *M. haemolytica* strains as antigen sources to further optimize and improve the specificity and sensitivity of the ELISA assay. In addition, future studies incorporating semi-quantitative analysis with characterized samples could provide additional information on the levels of antibody present in relation to the animals’ status, and potentially improve the diagnostic accuracy of the ELISA assay.

The IgG, IgM, and IgA isotype-specific *M. haemolytica* ELISAs developed in this study were aimed at improving the detection of *M. haemolytica* infections, and subsequent immune responses in livestock. The different isotypes of antibodies (IgG, IgM, and IgA) have distinct functions in the immune response, therefore testing for these specific isotypes can provide unique information about the stage and progression of the infection. That is, IgG is the most common class of antibody and is typically produced after either initial or subsequent infection. The IgG antibodies remain in the blood for a long period of time and can potentially provide protection against reinfection [24]. Additionally, the detection of IgG antibodies in serum can be used to diagnose a past infection with *M. haemolytica* and compare the ability of different vaccine formulations to induce a robust humoral response. IgM is the first immunoglobulin class produced after an infection and is usually detectable within the first week of infection. In general, the level of IgM peaks at the early stage of the infection then declines after class switching; therefore, measurement of IgG and IgM in the same animal at the same time could be used to differentiate recent versus past infection. Utilizing the same approach described elsewhere [73], combinations of different immunoglobulin-specific ELISAs can be used to differentiate early infection in newborns from maternally derived antibodies. As bovine colostrum has much lower concentrations of IgM and IgA (91.77 g/L IgG and approximately 5 g/L IgA and IgM), a combination of IgG, IgM, and IgA antibody levels can be used to determine whether antibodies in calves are likely maternally derived [74,75], or the result of an active infection. That is, high concentrations of IgG in neonatal serum indicate the successful passive transfer of maternal antibodies [41,76]. In contrast, the detection of elevated IgM antibody in calf serum indicates an early infection, regardless of the presence of maternally derived Ig, which is primarily IgG. As much of the IgA is produced by plasma cells under mucosal surfaces, levels of IgA-specific antibodies in serum could support a diagnosis of recent active infection. Therefore, the use of IgG, IgM, and IgA isotype-specific *M. haemolytica* ELISAs together could provide a more complete picture of whether the immune responses in calves were passively acquired from the dam, or actively generated by recent infection [25,32,33]. 

In this study, we aimed to detect the presence of antibodies to whole *M. haemolytica* cells rather than antibodies to leukotoxin specifically because the use of whole-cell antigen allows for the detection of a wider range of antibodies that might be more sensitive and specific than leukotoxin-specific assays. Nonetheless, future studies exploring the development of *M. haemolytica* leukotoxin-specific assays may provide a valuable addition to the serological toolbox for *M. haemolytica* infection and offer further insights into the protective immunity of cattle against the infection.

Amid the wide use of *M. haemolytica* vaccines in cattle, measurement of IgG and IgM antibody levels in a herd might be used to determine the effectiveness of vaccination programs. A herd that has been vaccinated against *M. haemolytica* should have high levels of IgG and IgM antibodies after the initial dose, confirming an active immune response. Although immunity against *M. haemolytica* involves several immune factors, antibodies detected by ELISAs can be one indicator to determine duration of the protective immunity in addition to other diagnostic assays and clinical status of the herd [57]. 

As *M. haemolytica* is an opportunistic bacterial species which resides in the upper respiratory tract of cattle, the bacteria may be isolated from healthy, clinically normal cattle [59]. Such cattle may have some baseline levels of *M. haemolytica*-specific antibodies, including vaccine-induced antibodies, which pose complications in the interpretation of serological assay results. The results in this study showed that IgG, IgM, and IgA antibody levels were elevated in animals which experienced *M. haemolytica* isolation. When used in combination with the IgM-specific *M. haemolytica* ELISA, the presence of high concentrations of these two antibody isotypes supports a likely active infection [57,58]. The result of these assays, along with the animal’s clinical status and other diagnostic test results, can provide valuable information about the host’s immune response to the infection. 

As the test performance of the ELISAs in this study was determined based on clinical specimens from a field experiment, a serological test using serum samples from cattle with negative culture results may lead to false negative results, as cattle with a negative culture may still have been exposed to the bacteria and have developed antibodies. This may explain the low diagnostic specificity of the tests at cutoffs that give high diagnostic sensitivity. On the other hand, although the diagnostic specificity of the IgA test was lower than the other isotypes in serum specimen, it still demonstrated a reasonable diagnostic sensitivity. IgA is specific to mucosal immunity, which can be used to predict protection against mucosal disease pathogen such as *M. haemolytica* [25,33]. These findings suggest that the *M. haemolytica* ELISAs can be adapted for the detection and quantification of antibodies in serum specimens, and support the use of these tests for disease surveillance and disease prevention research in feedlot cattle. In addition, understanding the potential for cross-reactivity with other bacteria is an important consideration for any diagnostic assay, and future studies should be conducted to further evaluate the specificity and cross-reactivity of the ELISA with other bacteria.

The results of this study indicate that the IgG ELISAs developed for the detection of *M. haemolytica* antibodies in feedlot cattle are reliable and have a high diagnostic sensitivity, specificity, and results that relate to the isolation of the bacteria from affected cattle. This suggests that these tests can be useful in identifying *M. haemolytica* infections in this population of cattle. However, it is important to note that inadequate overall host immunity plays a role in the development of *M. haemolytica*-associated disease, particularly in young calves [10]. Thus, high *M. haemolytica*-specific antibody levels may not necessarily indicate disease, but rather reflect a protected state. It is important to note that the detection of antibodies alone is not sufficient to confirm a diagnosis of *M. haemolytica* infection. A definitive diagnosis should be based on a combination of diagnostic methods, including bacteriological culture, PCR, clinical findings, and vaccination history. This approach will provide a more comprehensive understanding of the infection status and aid in the development of appropriate treatment and management strategies.

In conclusion, this assay allows for the detection of IgG, IgM, and IgA antibodies generated in response to *M. haemolytica,* which can help measure isotype specific immune responses to this important BRD pathogen. Such testing can provide important information about the protective mechanisms of the immune system and can be used to develop new vaccines and diagnostic tools. This is especially important to help understand the host immune response to opportunistic pathogens that are present in normal healthy animals, but may also cause BRD. The systematic study of the levels of these antibodies over time can be used to understand the dynamics of the infection and allow for risk assessment or risk-based interventions that may decrease the risk of BRD. More research is needed to improve understanding of *M. haemolytica* disease ecology and effective diagnostic and control strategies, and these assays may help inform these strategies. 

## Figures and Tables

**Figure 1 animals-13-01531-f001:**
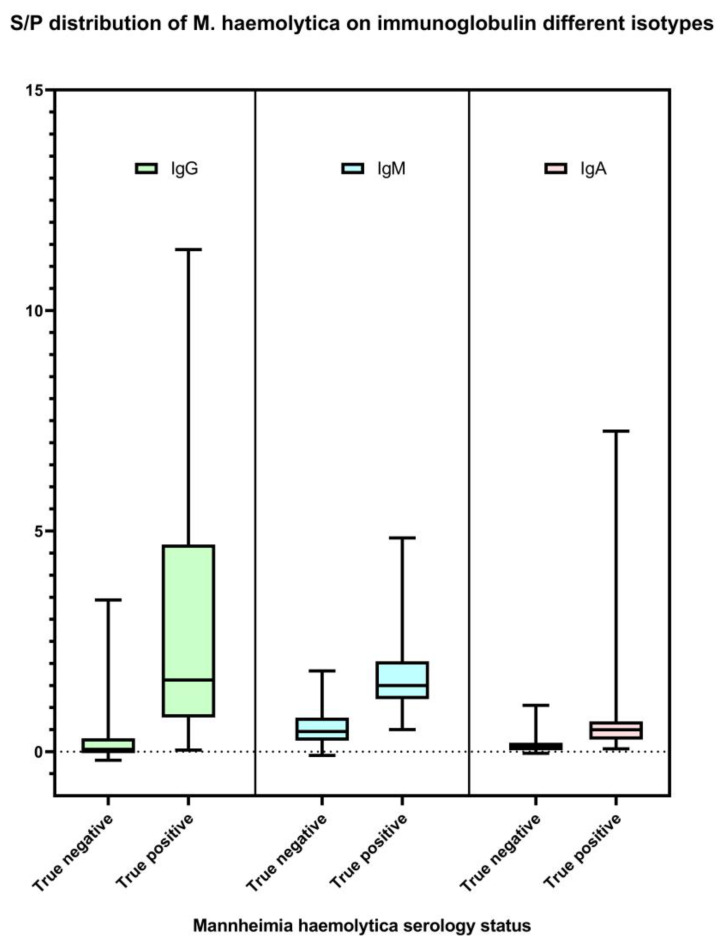
Distribution of *M. haemolytica* IgG, IgM, and IgA isotype-specific ELISA sample-to-positive (S/P; *Y*-axis) based on test validation samples (*n* = 145). Serum samples collected from the cattle that were negative for *M. haemolytica* culture on study day 0 (*n* = 100) were classified as true *M. haemolytica* antibody-negative. Serum samples collected from the cattle at least 14 days after *M. haemolytica* isolation positive (*n* = 45) were classified as true *M. haemolytica* antibody positive.

**Figure 2 animals-13-01531-f002:**
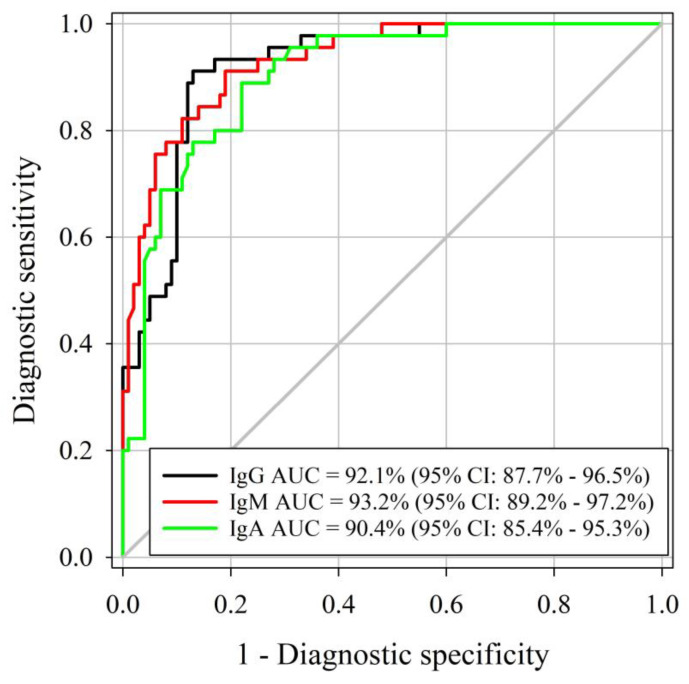
Receiver operating characteristic (ROC) curve analysis of the *M. haemolytica* IgG, IgM, and IgA isotype-specific ELISA sample-to-positive (S/P) based on testing serum samples from *M. haemolytica*-culture-free (true negative) (*n* = 100) and *M. haemolytica*-positive (true-positive) cattle (*n* = 45).

**Table 1 animals-13-01531-t001:** Inter-assay and intra-assay repeatability of the *Mannheimia haemolytica* IgG, IgM, and IgA isotype-specific ELISAs.

		Inter-Assay Repeatability (30 Samples, 4 Rounds)	Intra-Assay Repeatability (8 Samples, 12 Rounds)
		ICC (95% CI)	ICC (95% CI)
Antibody isotype	IgG	97.6% (95.5%, 98.8%)	99.9% (99.8%, 100.0%)
	IgM	87.2% (77.7%, 93.3%)	99.9% (99.7%, 100.0%)
	IgA	98.6% (97.6%, 99.3%)	99.6% (99.1%, 99.9%)

**Table 2 animals-13-01531-t002:** *M. haemolytica* IgG, IgM, and IgA ELISA diagnostic sensitivity and specificity by cutoff.

	IgG		IgM		IgA	
Cutoff (S/P)	Diagnostic Sensitivity (95% CI)	Diagnostic Specificity (95% CI)	Diagnostic Sensitivity (95% CI)	Diagnostic Specificity (95% CI)	Diagnostic Sensitivity (95% CI)	Diagnostic Specificity (95% CI)
0.1	97.8 (93.3, 100.0)	59.0 (49.0, 69.0)	100.0 (100.0, 100.0)	6.0 (2.0, 11.0)	97.8 (93.3, 100.0)	50.0 (40.0, 60.0)
0.2	95.6 (88.9, 100.0)	67.0 (58.0, 76.0)	100.0 (100.0, 100.0)	20.0 (13.0, 28.0)	88.9 (77.8, 97.8)	74.0 (65.0, 82.0)
0.3	93.3 (84.4, 100.0)	75.0 (66.0, 83.0)	100.0 (100.0, 100.0)	31.0 (22.0, 40.0)	68.9 (55.6, 82.2)	89.0 (82.0, 95.0)
0.4	91.1 (82.2, 97.8)	85.0 (78.0, 91.0)	100.0 (100.0, 100.0)	45.0 (36.0, 55.0)	57.8 (44.4, 71.1)	94.0 (89.0, 98.0)
0.5	91.1 (82.2, 97.8)	87.0 (80.0, 93.0)	100.0 (100.0, 100.0)	52.0 (42.0, 62.0)	46.7 (33.3, 62.2)	96.0 (92.0, 99.0)
0.6	86.7 (75.6, 95.6)	88.0 (81.0, 94.0)	97.8 (93.3, 100.0)	61.0 (51.0, 70.0)	35.6 (22.2, 48.9)	96.0 (92.0, 99.0)
0.7	77.8 (64.4, 88.9)	90.0 (84.0, 95.0)	93.3 (84.4, 100.0)	66.0 (57.0, 75.0)	24.4 (13.3, 37.8)	96.0 (92.0, 99.0)
0.8	73.3 (60.0, 86.7)	90.0 (84.0, 95.0)	91.1 (82.2, 97.8)	77.0 (69.0, 85.0)	20.0 (8.9, 31.1)	99.0 (97.0, 100.0)
0.9	73.3 (60.0, 86.7)	90.0 (84.0, 95.0)	84.4 (73.3, 93.3)	82.0 (74.0, 89.0)	20.0 (8.9, 31.1)	99.0 (97.0, 100.0)
1.0	66.7 (53.3, 80.0)	90.0 (84.0, 95.0)	82.2 (71.1, 93.3)	89.0 (82.0, 95.0)	20.0 (8.9, 31.1)	99.0 (97.0, 100.0)

## Data Availability

Not applicable.

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
