# Peer review of "Detection of Mannheimia haemolytica-Specific IgG, IgM and IgA in Sera and Their Relationship to Respiratory Disease in Cattle"

_animals, 2023, doi:10.3390/ani13091531_

Round 1
Reviewer 1 Report
This is a well written and described study. I had few edits or concerns with the review of the manuscript:
Is "Mh-free" possible given the commensal nature of Mh and limitations with culture techniques? This limitation is described well in discussion, but perhaps a different term should be used?
Is it typical to use only a single antigen type to validate ELISA? Would testing with another Mh strain allow better insight into the most suitable antigen candidate?
Author Response
Thank you for taking the time to review our manuscript and for providing us with valuable feedback. Please find our response listed below.
Reviewer 1:
Is "Mh-free" possible given the commensal nature of Mh and limitations with culture techniques? This limitation is described well in discussion, but perhaps a different term should be used?
Thank you for your insightful comments on our manuscript. We appreciate your feedback and have carefully considered your suggestion regarding the term "Mh-free." We agree that due to the commensal nature of Mannheimia haemolytica and the limitations of current culture techniques, it is difficult to claim that an animal is completely free of this bacterium.
To address this concern, we used the term "Mh-culture free” to describe the absence of clinical signs and culturable levels of M. haemolytica in our study animals. We believe that this term accurately reflects the limitations of our detection methods and the fact that M. haemolytica may still be present at unculturable amounts in subclinical animals.
Is it typical to use only a single antigen type to validate ELISA? Would testing with another Mh strain allow better insight into the most suitable antigen candidate?
Thank you for your insightful comments on our manuscript. With regard to your question about using a single antigen type to validate the ELISA, we chose to utilize a serotype A1 M. haemolytica strain as the antigen source for several reasons.
Firstly, serotype A1 strains have been shown to be more frequently associated with disease, and therefore likely to have enhanced utility for an immunoassay, as described in Lines 417-420 of the manuscript. Secondly, the serotype A1 strain we used has been characterized and well-studied in previous research, and is available through culture collections, which allowed for greater control and reproducibility of our assay.
While it may be possible to use other M. haemolytica strains to validate the ELISA, we believe that our approach provides a solid foundation for further studies and allows for direct comparison with previous research that has utilized the same antigen source (Lines 504-515). Nonetheless, we acknowledge that future studies utilizing additional M. haemolytica strains could potentially provide further insight into the most suitable antigen candidate as described in Lines 421-423.
Reviewer 2 Report
Authors have developed the ELISA for the detection of Mannheimia haemolytica that to three different antibody based.
The current manuscript is well written with acceptable diagnostic parameters.
Following queries need to be address:
1) Did author have study the cross reactivity of the assay with same genus and different species of bacterium? and with similar class/group of bacterium ? Did author have conducted any experiment to find assay's cross reactivity with non-targeted bacteria ? if it so, please let add the data.
2) The cutoff is 0.8, it is too low for diagnostic test, however, acceptable with control set up and SOP. Validation with samples seems good with this cut off. Authors are requested to address it well in discussion part.
3) if possible, supplymentary data can be provided if any!
4) Did author try to make semi-quantification ELISA based on OD during the experiment ?
5) Did you mention total ELISA step's time duration (particularly total incubation time and approx complete ELISA time)? if so please let me know if I missed in manuscript. many commercial kits development process are also considering time duration feasibility.
6) Conclusive paragraph needs improvement specifically write on diagnostic improvement.
Author Response
Thank you for your thoughtful review of our manuscript. We really appreciate the time you took to provide detailed feedback. Your comments have been extremely helpful in improving the clarity and overall quality of our work. Our responses and the edits are listed below.
Reviewer 2:
1) Did author have study the cross reactivity of the assay with same genus and different species of bacterium? and with similar class/group of bacterium ? Did author have conducted any experiment to find assay's cross reactivity with non-targeted bacteria ? if it so, please let add the data.
Thank you for your comments on our manuscript. With regards to your question about the cross-reactivity of our assay with bacteria from the same genus, different species, or similar class/group, we did not conduct any experiments to evaluate this.
Our primary objective in this study was to evaluate the potential of the ELISA as a diagnostic tool for detecting antibodies against M. haemolytica. While cross-reactivity with other bacteria is an important consideration for the diagnostic assay, we did not have the resources or scope in our current study to explore this.
Nonetheless, we acknowledge the importance of understanding the potential for cross-reactivity in any diagnostic assay, and we made note of this limitation in our manuscript (Lines 487-490).
2) The cutoff is 0.8, it is too low for diagnostic test, however, acceptable with control set up and SOP. Validation with samples seems good with this cut off. Authors are requested to address it well in discussion part.
Thank you for your comments on our manuscript. We appreciate your feedback regarding the cutoff value of 0.8 that we used in our study.
We agree that this cutoff value may not be ideal for a diagnostic test, but we would like to emphasize that we chose this value based on the best balance between diagnostic sensitivity and diagnostic specificity. Our validation with samples demonstrated good performance with this cutoff value, and we believe that it provides a useful starting point for further optimization and development of the assay.
Furthermore, we provided test performance at different cutoff values in Table 2 of the manuscript, which can be used by readers to select a cutoff value that aligns with their specific diagnostic goals. By providing this information, we hope to facilitate the use of our assay in a range of diagnostic applications.
3) if possible, supplementary data can be provided if any!
We agree that providing supplementary data would be a valuable addition to our manuscript. With our revised submission, we have included a supplementary table containing information on the animals used in the study including their ID numbers, M. haemolytica culture results, status, and IgG, IgM, and IgA S/P ratio values. Thank you for the suggestion.
4) Did author try to make semi-quantification ELISA based on OD during the experiment?
With regards to your question about semi-quantification of our ELISA assay based on OD measurements, we did not perform this type of analysis in our study. With limitations on characterizing animals' status and limited number of animals in the study, we could not summarize additional information on the levels of antibody present in the samples in relation to the status of the animals which is needed for the semi-quantification analyses. Our primary objective of this study was to evaluate the potential of the ELISA as a diagnostic tool for detecting M. haemolytica-specific antibodies in serum samples. Nonetheless, we appreciate your suggestion and will consider incorporating semi-quantitative analysis into future studies. We discussed the possibility and potential benefits of incorporating this type of analysis in the discussion section of our manuscript (Lines 421-423).
5) Did you mention total ELISA step's time duration (particularly total incubation time and approx complete ELISA time)? if so please let me know if I missed in manuscript. many commercial kits development process are also considering time duration feasibility.
Upon review, we have realized that we did not explicitly mention the total incubation time or the approximate complete ELISA time in our manuscript. We appreciate your feedback on the importance of this information for the feasibility of commercial kit development.
We have revised the Materials and Methods section of our manuscript to include the total incubation time and approximate complete ELISA time for our assay (Lines 227-228). We hope that this information will be useful for readers and commercial developers interested in our assay.
6) Conclusive paragraph needs improvement specifically write on diagnostic improvement.
Thank you for the suggestion. We have revised the conclusive paragraph in the discussion section incorporated language specifically addressing diagnostic improvement and highlighted the importance of measuring isotype-specific immune responses to M. haemolytica. Furthermore, we have emphasized the need for continued research to improve our understanding of the disease ecology of M. haemolytica and develop more effective diagnostic and control strategies (Lines 504-515).